# The receptor for advanced glycation endproducts (RAGE) modulates T cell signaling

**James C. Reed**[1,2¤‡]**, Paula Preston-Hurlburt**[2‡]**, William Philbrick**[3]**, Gabriel Betancur**[2]**, Maria Korah**[2]**, Carrie Lucas**[2]**, Kevan C. Herold**[2,3] *****

**1** Howard Hughes Medical Institute, Yale School of Medicine, New Haven, CT, United States of America,
**2** Department of Immunobiology, Yale School of Medicine, New Haven, CT, United States of America,
**3** Department of Internal Medicine, Section of Endocrinology, Yale School of Medicine, New Haven, CT, United States of America

¤ Current address: Department of Pediatrics, Mount Sinai Hospital, New York, NY, United States of America
‡ These authors share first authorship on this work.
* kevan.herold@yale.edu

**Data Availability Statement:** All relevant data are within the manuscript and its Supporting Information files.

## Abstract

The receptor for advanced glycation endproducts (RAGE) is expressed in T cells after activation with antigen and is constitutively expressed in T cells from patients at-risk for and with type 1 diabetes mellitus (T1D). RAGE expression was associated with an activated T cell phenotype, leading us to examine whether RAGE is involved in T cell signaling. In primary CD4+ and CD8+ T cells from patients with T1D or healthy control subjects, RAGE- cells showed reduced phosphorylation of Erk. To study T cell receptor signaling in RAGE+ or–T cells, we compared signaling in RAGE+/+ Jurkat cells, Jurkat cells with RAGE eliminated by CRISPR/Cas9, or silenced with siRNA. In RAGE KO Jurkat cells, there was reduced phosphorylation of Zap70, Erk and MEK, but not Lck or CD3ξ. RAGE KO cells produced less IL-2 when activated with anti-CD3 +/- anti-CD28. Stimulation with PMA restored signaling and (with ionomycin) IL-2 production. Silencing RAGE with siRNA also decreased signaling. Our studies show that RAGE expression in human T cells is associated with an activated signaling cascade. These findings suggest a link between inflammatory products that are found in patients with diabetes, other autoimmune diseases, and inflammation that may enhance T cell reactivity.

## Introduction

The receptor for advanced glycation endproducts (RAGE) is a type 1 transmembrane pattern recognition receptor of the immunoglobulin superfamily that serves as a receptor for non-enzymatically glycosylated endproducts. These and other ligands, including damage-associated molecular patterns (DAMPs) [1–6], such as S100b/calgranulins, amyloid, HMGB1, are increased in patients with diabetes as well as other autoimmune conditions and inflammatory states [7]. Other ligands, such as RNA and DNA that are released from dying cells [6] also bind

**Funding:** The work was supported by grant R01DK057846 from the NIH and 201300688 from the Juvenile Diabetes Research Foundation to KCH and a fellowship award from Howard Hughes Medical Institute to JCR.

**Competing interests:** The authors have declared that no competing interests exist.

and activate RAGE. RAGE is thought to be involved in the pathogenesis of end organ complications of diabetes [8]. In addition, the increased cellular expression of RAGE and availability of ligands has suggested a role in inflammatory reactions, possibly through its ability to bind and activate antigen presenting cells independently of TLR9 [6, 9–11]. For example, in anti-phospholipid syndrome, increased levels of HMGB1 and soluble RAGE correlate with clinical manifestations of the disease [7]. Curiously, deletion of RAGE exacerbates lymphoproliferative syndrome and lupus nephritis in B6-MRL Fas lpr/j mice, thought to be due to reduced apoptosis in RAGE deficient T cells [12]. The levels of RAGE expression are higher in T cells from patients with type 1 diabetes (T1D) and in others with hyperglycemia, and increased RAGE expression in T cells is associated with progression to T1D in at-risk relatives [10, 13].

We have shown that RAGE is involved in adaptive T cell responses in murine models. RAGE deficiency or blockade affected murine T cell activation and differentiation that were needed for autoimmune diabetes in NOD mice and antigen recognition in ovalbumin-reactive CD4+ T cells [2, 14–16]. RAGE blockade or deficiency prevented transfer of autoimmune diabetes or islet graft rejection in diabetic NOD mice [2, 14]. In addition, we found that RAGE was involved in development of pathogenic Th1 responses since RAGE-/- OTII T cells did not proliferate and produced less IFNγ when they were transferred into wild-type or RAGE deficient recipient mice expressing ovalbumin [15]. In other studies, we showed that RAGE-deficient mice were also protected from OVA-induced pulmonary disease [16].

RAGE is also involved in human T cell responses. It is expressed on viral antigen (EBV) reactive human CD8+ T cells and increased in expression when they are triggered by antigen [10]. We found constitutive RAGE expression is higher in T cells in studies of peripheral blood mononuclear cells (PBMC) from more than 25 patients with established Type 1 diabetes (T1D) [10, 13]. We also previous studied 22 relatives of patients with T1D who were at-risk for disease and found increased expression of RAGE in T cells, compared to 18 healthy control subjects, particularly among those who progressed to T1D [13]. RAGE+ T cells were increased in relatives at-risk for T1D before they presented with clinical disease. Transcriptional and phenotypic studies showed that the RAGE+ CD4+ and CD8+ T cells had a memory phenotype and also had higher levels of expression of EOMES, KLRG1, IRF4, CXCR3 and constitutive expression of STAT3 indicating an activated phenotype [13]. Curiously, the RAGE+ cells had increased rates of survival, similar to autoreactive T cells in MRL mice. These findings suggested an interaction of T cells with environmental factors that could enhance their pathogenicity through binding of ligands to RAGE [13]. However, it was not clear how RAGE affected T cell signaling and how this contributes to the inflammatory profile of RAGE+ T cells. We postulated that the activated phenotype of RAGE + cells may affect T cell receptor signaling and therefore studied the role of RAGE in T cell activation events.

RAGE signaling among different cells utilizes different cellular pathways [17]. In a variety of mammalian cell systems including human lung, RAGE cytoplasmic domain was found to interact with Dia-1 to transduce extracellular environmental cues evoked by binding of RAGE ligands to the receptor, a consequence of which is Rac-1 as well as Erk1/2 activation [18]. Based on the previous studies we postulated that RAGE may affect T cell signaling. Unlike murine RAGE+ T cells, the expression of RAGE in human T cells was intracellular and was associated with the endosomes [10]. Thus, the role of RAGE among activation pathways in human T cells is likely to be different from other cells on which the expression is on the cell surface. We therefore analyzed the role of RAGE in signaling among primary human T cells and Jurkat cells.

## Methods

### Cell lines and human samples

Peripheral blood mononuclear cells (PBMC) were obtained and frozen from 8 patients with T1D of > 2 yrs duration and 6 healthy control adult subjects without a history of T1D. Blood samples for the healthy control subjects were purchased from the New York Blood Center. The patients with T1D (2 M, 6 F), age (mean SD) 25.3 (16.5) yrs, had a history of T1D for 10.5 (13) yrs (range 1–39 yrs). PBMC were purified by Ficoll centrifugation. The cells were harvested from the interface, washed and counted and frozen at $10^7$/ml in freezing media (FBS, 10%DMSO). After overnight at -80°C, the cells were transferred to liquid nitrogen storage.

The Jurkat cells used were of the ATCC E6.1 line (https://www.atcc.org/products/all/TIB-152.aspx). Upon receipt, the line was expanded and frozen. The cells used for experiments were used within 1 month of thawing vials from the originally expanded line. Written informed consent was obtained from cell donors. The study was approved by the Yale University Institutional Review Board (FWA#00002571).

### Cell signaling and FACS

Frozen vials of patient samples or healthy control samples were thawed into complete RPMI containing 5% FBS at 37°C and washed once and counted. The cells were plated at 1 x10^6 cells/ml and stimulated using 0.3μg/ml of anti-CD3 (OKT3) and anti-CD28 for 3 days. The cells were washed on day 4, and placed back into culture (1 x10^6 cells/ml) with hIL2 (100 IU/ml) for another 3 days. The cells were harvested on Day 7 for signaling studies.

Resting cells were harvested and washed and counted and resuspended at 1 x 10^6/ml in RPMI 5% media without IL2. They were placed on ice for 20 min after which time anti-human IgG was added (0.2μg/ml Affinipure F(ab')2 goat anti human IgG (Jackson Immuno Research)) and cultured for another 30min. They were then cultured with anti-CD3 mAb (0.2μg/ml OKT3) on ice for 15min after which the tubes were placed in a 37°C water bath for 1, 2, 4, 5, or 10 min. At each time point tubes were removed from water bath and placed into ice to stop the reaction. For cytokine release, the cells were stimulated with PMA (400 nM) and the supernatants were harvested the following day.

For intracellular staining, the cells were washed once in ice cold PBS and fixed and permeabilized according to the manufacturer's instructions (BD Phosflow Fix 1 buffer (BD Biosciences, San Jose, CA, Cat#557870) and BDPhosflowPerm III, Cat #558050) according to the manufacturer's instructions. After washing the cells were stained for CD4, CD8, RAGE, and signaling molecules and analyzed. The antibodies used for flow cytometry were: AGER-AF488 (Bioss, Woburn, MA@1:10 dilution, Cat# bs-4999R-A488), and, from BD Biosciences (San Jose, CA): PerCP-Cy5.5 anti-phospho-ZAP70 (pY319/Syk(Y352)(17A/pZAP70, 5 μl, Cat# 561459), Alexa Fluor® 647 Mouse anti-MEK1 (pS218)/MEK2 (pS222) (024–836, 5 μl, Cat#562460), Alexa Fluor 647 anti-ERK ½ (pT202/pY204) (20A, 20 μl, Cat#612593), Alexa Fluor® 647 Mouse anti-CD247 (pY142)(K25-407.69, 20 μl, Cat#558489), Alexa Fluor® 647 Mouse anti-MEK1 (Total MEK) (25/MEK1, 20 μl, Cat#560101), from Biolegend (San Diego CA): Pacific Blue-anti-human CD4 (RPAT4, 5 μl, Cat#300524), BV650- anti-human CD8 (RPAT8, 5 μl, Cat#301042), Phycoerythrin (PE) anti-Lck Phospho (Tyr394) Antibody (A18002D, 5 μl, Cat#933104), PE anti-ZAP70 Antibody (Total zap70) (A15114B, 5μl, Cat#691204), Alexa Fluor® 647 anti-Lck Antibody (Total LCK) (LCK-01, 5 μl, Cat#628304), Alexa Fluor® 594 anti-ERK1/2 Antibody (Total ERK) (W15133B, 1 μl, Cat#686903), and AF647-pLCK(Y394) from Novus (1μl, discontinued). The FACS analysis was done on a BD LSRFortessa.

## CRISPR/Cas9 RAGE knockouts

CRISPR/Cas9 knockouts were produced using the pSpCas9(BB)-2A-Puro (PX459) V2.0 expression vector, a gift from Feng Zhang [19]. Guide sequences targeting the 5' and 3' ends of the RAGE gene were selected using the Broad Institute algorithm (https://portals.broad institute.org/gpp/public/analysis-tools/sgrna-design [20] (5′: GTGGCTCACCCCACAGA CTG-3′, 3′: TTATTTAGTGGGAGCCCCAG-3′. The excision generated a 2475 bp deletion eliminating the bulk of the coding sequences. pMAX GFP plasmids from the Amaxa Nucleofector 2b set were used as transfection controls. The cells were cloned at limiting dilution and RAGE -/- clones were identified by the absence of full length RAGE DNA and presence of the deleted DNA by PCR and FACS (S1 Fig).

We also assessed RAGE signaling by Western blots in Jurkat cells transfected with control or RAGE siRNA. RAGE gene expression was silenced in Jurkat cells with a mixture of 3 Silencer Select siRNAs s1166, s1167, and s1168 (ThermoFisher Scientific, Waltham, MA), at a final total siRNA concentration of 0.5 μM. Control transfection solutions were made of 0.5 μM Silencer™ Negative Control No.1 siRNA (AM4611) and Silencer™ Cy3™-labeled Negative Control No. 1 siRNA (ID AM4621). Standard techniques were used for transfection. Silencing was verified by FACS analysis and was found to be a 75–80% reduction in the mean fluorescence intensity (MFI) of RAGE vs control siRNA transfected cells.

The siRNA-treated Jurkat cells were stimulated with cross linked anti-CD3 mAb for the indicated periods of time and then flooded with ice-cold PBS. Digests of the cells were prepared with Cell Lysis Buffer with Halt Protease and Phosphatase inhibitors (ThermoFisher Scientific) and frozen. For western blotting, the lysates were thawed and loaded onto a 4–12% Bis-Tris gel (NuPAGE Novex). The gels were transferred to 0.45 μm nitrocellulose pre-cut blotting membranes and probed with anti-B-Tubulin (9F3), Phospho-ZAP-70 (Tyr319/Syk (Tyr352) (65E4), and Phospho-p44/42 MAPK (ERK1/2) (Thr202/Tyr204) (D13.14.4E) and secondary antibodies (all Cell Signaling Technologies, Danvers, MA). Membranes were stripped and reprobed with rabbit IgG anti-B-Tubulin (9F3), Diap1, ZAP-70 (99F2), and p44/42 MAPK (ERK1/2) (137F5) (all Cell Signaling Technologies). Developed membranes were read on a Bio-Rad ChemiDoc.

## Cytokine production

WT or RAGE-/- Jurkat cells were stimulated for 24 hrs with cross-linked OKT3 or OKT3/ CD28 mAbs (1 μg/ml) or phorbol 12-myristate 13-acetate (PMA, 400 nM) and ionomycin (500 ng/mL) The levels of IL-2 in supernatants after 24 hrs were measured by Luminex assay (ThermoFisher).

## Statistics

All analyses were done with Graph Pad Prism 7.01 software. For comparisons of activation, the AUC of the signals (for flow MFI) was calculated using the trapezoidal rule. The differences between groups were compared by t-test, for normally distributed, or Mann-Whitney tests for data that were not normally distributed. When multiple comparisons were done, an ANOVA (one or two way) was done as detailed. Unless indicated, the mean±SEM is shown.

# Results

## Signaling in RAGE+ and—human T cells

Increased expression of intracellular RAGE in T cells is a feature of patients with T1D and at-risk subjects who progress to overt disease [13]. Both RAGE+ and RAGE- T cells can be

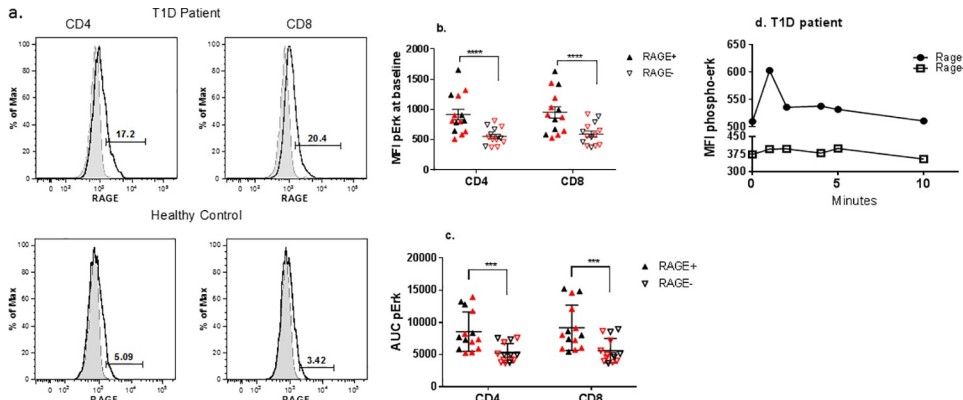

**Fig 1. Signaling in RAGE+ and RAGE⁻ primary T cells from T1D patients and healthy control subjects.** (a) Intracellular RAGE staining in CD4+ and CD8+ T cells from a representative patient with T1D (top) and a healthy control subject (bottom). Shaded histograms are staining with the isotype control. Solid lines are staining with anti-RAGE antibody. (b) The MFI of pErk in unstimulated CD4+ and CD8+ T cells (identified by FACS staining) from healthy control subjects and patients with T1D. Each symbol represents a cell donor. Red symbols indicate cells from patients with T1D (n = 8), black symbols represent healthy control subjects (n = 6). Filled symbols are RAGE+ and open symbols are RAGE- cells. (****p<0.001, two way ANOVA for paired samples of RAGE+ and RAGE- in each subject with Sidak's multiple comparison). (c) The AUC of pErk MFI in RAGE+ and RAGE- CD4+ and CD8+ T cells that were activated with cross-linked anti-CD3 mAb. Each symbol represents a cell donor as indicated in b. (Two way ANOVA for paired samples (RAGE+ and RAGE-) with Sidak's multiple comparison, ***p<0.001). (d) A representative example of the phosphoflow Erk1 response used to generate the data shown in (b) and (c) is shown for CD4+ cells from a patient with T1D.

identified by flow cytometry. To understand the significance of RAGE expression on activation of these T cells, we studied T cells from healthy control subjects (n = 6) and patients with T1D (n = 8) that were RAGE positive or negative by FACS staining, that were activated with anti-CD3 mAb. Similar to our previously reported findings, the overall levels of RAGE expression were higher in the patients with T1D (in CD4+ cells 7.5±1.65% in T1D vs 5.0±0.95% in healthy controls; in CD8+ cells, 10.4±2.92% vs 5.77±0.93%, Fig 1A), but the levels were similar after the cells had been cultured in IL-2 prior to the phospho-flow studes. Under basal conditions, the levels of pErk were lower in CD4+ and CD8+ RAGE- T cells (Fig 1B, ****p<0.0001). Following activation with cross linked anti-CD3 mAb, the phosphorylation of Erk was lower in the RAGE–vs RAGE+ CD4+ and CD8+ T cells from T1D patients and healthy control subjects (****p<0.0001) (Fig 1C and 1D). We did not find a difference in the levels of pErk in RAGE + or RAGE- T cells from patients with T1D or healthy control subjects.

## T cell activation in RAGE+ and RAGE-deficient T cells

To determine the mechanisms of RAGE effects on T cell signaling, we prepared RAGE KO cells by deleting the RAGE gene using Crispr/Cas9 in Jurkat cells (S1 Fig). We analyzed TCR signaling in the cells by phospho-flow cytometry following stimulation with cross-linked anti-CD3 mAb. Prior to activation, the levels of total Erk and Lck were higher in RAGE+/+ vs RAGE KO Jurkat cells, but the ratio of phospho:total ERK, Lck, and CD3ξ chain were similar across cells (Fig 2). These did not reflect a general functional impairment of RAGE KO cells since there were similar levels of proliferation compared to the WT cells (not shown). In addition, a greater proportion of the RAGE KO cells were CD3+ (***p< 0.001) but the MFI of CD3 on their cell surfaces was lower consistent with the reduced expression of signaling molecules in RAGE-deficient T cells (**p<0.01) (S2 Fig).

When the KO cells were activated with cross-linked anti-CD3 mAb, we found reduced phosphorylation of Erk (p = 0.02) and Zap70 (p = 0.03) but not Lck (p = 0.58) (Fig 2). To

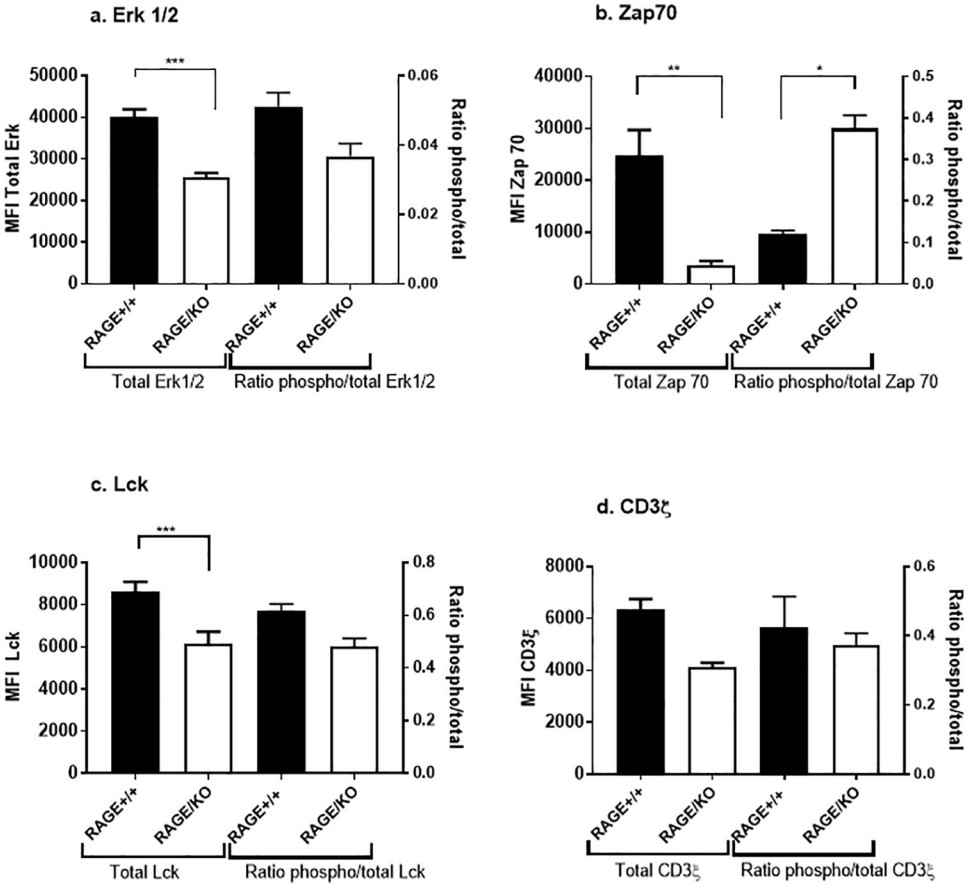

**Fig 2. Expression of signaling molecules in WT RAGE+/+ and RAGE/KO Jurkat cells.** The mean fluorescence intensity (MFI) of total Erk (a), Zap70 (b), Lck (c), and CD3ξ (d) (Left Y axis), and the ratio of phosphorylated to total molecules (Right Y axis) is shown. Comparisons were made between RAGE++ Jurkat cells and RAGE knockout Jurkat cells (RAGE/KO). The expression of total Erk1/2 and Lck were reduced in the KO cells (a) (n = 5, 5, 3, 5). The ratio of phosphorylated Zap 70/total Zap 70 was increased in the RAGE/KO cells because of reduced total Zap 70 expression. (ANOVA with Tukey's multiple comparison: *p<0.05, **p<0.01, ***p<0.001).

further explore this impairment in signaling, we analyzed phosphorylation of CD3ξ and MEK1/2, signaling steps prior to and after Erk and Zap70 signaling. CD3ξ phosphorylation was not significantly reduced in the absence of RAGE (p = 0.06) but MEK1/2 phosphorylation was reduced(p<0.05) (Fig 3A–3E).

These findings indicate that there is reduced phosphorylation of critical T cell signaling molecules when RAGE is deleted or even in normal T cells that lack RAGE expression. The reduced phosphorylation and signaling that we found in the RAGE-/- cells was not the result of overall reduction in the ligands because the ratio of phosphorylated:total Erk was reduced throughout the stimulation period even when corrected for the total Erk staining whereas phosphorylated CD3ξ was not reduced after correction for total CD3ξ staining (not shown). When the WT and RAGE/KO Jurkat cells were activated with PMA, which is an allosteric activator of protein kinase-C and bypasses Zap70 signaling, the induction of pErk and pMEK1/2 was similar despite lower levels in unstimulated cells (p = 0.01) (Fig 3F).

To determine whether these signaling events had functional consequences for the T cells, we compared the levels of IL-2 in the supernatants of the cultures after Jurkat cells were activated with anti-CD3 mAb with or without anti-CD28 mAb or with PMA+ionomycin. There

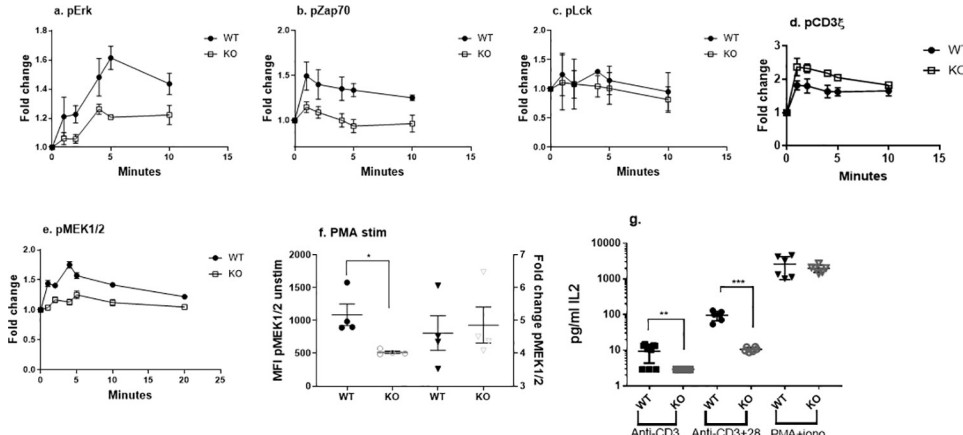

**Fig 3. Early signaling events in WT and RAGE KO Jurkat cells.** The levels of (a) p-ERK, (b) p-Zap 70, (c) p-Lck, (d) pCD3ξ and (e) pMEK were measured after stimulation of RAGE+/+ (filled circles) and RAGE/KO (open squares) Jurkat cells. There was a significant reduction in the average AUC of pErk (n = 3 each, Student's t-test p = 0.02) in the RAGE/KO Jurkat cells after stimulation with crosslinked anti-CD3 mAb, (b) p-ZAP70 (n = 3 each, Student's t-test p = 0.03), (c) pLck (n = 3, p = ns), (d) pCD3ξ (n = 3, p = ns), and (e) pMEK (n = 2). (f) The expression of pMEK1/2 before (L Y axis, Mann Whitney, p = 0.029) and the peak fold change (R Y axis, ns) after stimulation with PMA for 20 minutes in WT (filled symbols) and RAGE/KO (open symbols) Jurkat cells is shown. (g) RAGE+/+ (filled symbols) or RAGE/KO (open symbols) Jurkat cells were stimulated with anti-CD3 (n = 3), anti-CD3/28 mAbs(n = 3), or PMA +ionomycin (n = 4) for 24 hours and the levels of IL-2 were measured in supernatants. (ANOVA with post-hoc comparison **p<0.001, ***p<0.001).

were reduced levels of IL-2 after 24 hrs with anti-CD3 +/- anti-CD28 mAb from the KO cell cultures but the levels were similar with PMA/Ionomycin (Fig 3G). Collectively these findings suggested a role of RAGE in the earliest steps in CD3 activation and is associated with functional consequences for T cells.

To confirm these findings, we transfected Jurkat cells with pooled siRNA constructs to inhibit RAGE expression and studied activation by Western blot. We were able to achieve a 75–80% knockdown of RAGE, 48 hours after transfection, based on the MFI of RAGE staining, measured by FACS. There was a reduction in the phosphorylation of Zap70 in the Jurkat cells transfected with RAGE siRNA (two-way ANOVA, p = 0.0002, Fig 4).

## Discussion

These studies show that RAGE can modulate cell activation in human T cells. It is involved in T cell signaling at steps beyond the phosphorylation of the CD3ξ chain–downstream events including phosphorylation of Zap70, Erk1/2, and MEK are all affected. Bypassing pZap70 and CD3ξ with PMA restores cell signaling and IL-2 production. The interactions of RAGE are specific in the signaling cascade since phosphorylation of CD3ξ and Lck were unaffected. These findings suggest that RAGE-mediated modifications of T cell signaling cascade may be an important mechanism whereby T cell responses are modulated in response to inflammatory mediators.

RAGE expression is constitutive on innate immune cells and dynamic in T cells. It is found at low levels in resting T cells and increases when they are activated particularly in the presence of RAGE ligands [10]. We cannot be certain whether the increased levels of RAGE expression in T cells from patients with diabetes is the cause or result of T cell activation since hyperglycemia and other features of the diabetic environment lead to the generation of RAGE ligands. Nonetheless, our findings suggest a mechanism whereby environmental factors in diabetes may lead to the activation of adaptive and innate immune responses consistent with the

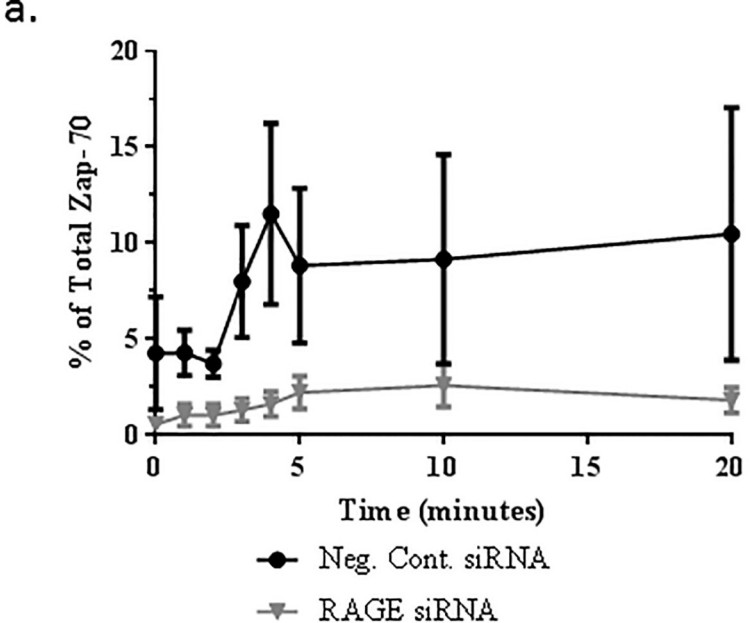

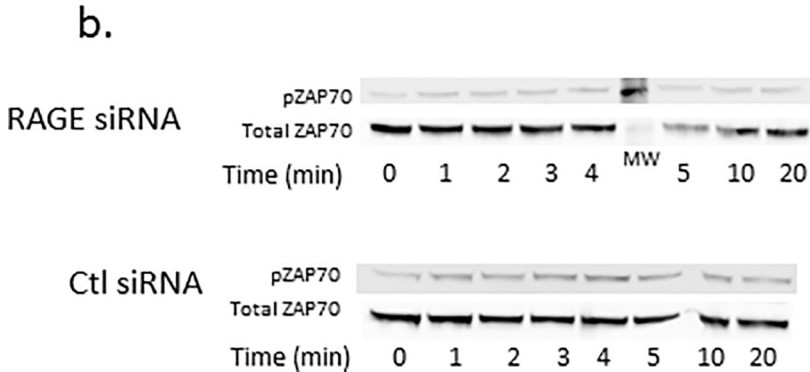

**Fig 4. Inhibition of early activation events in Jurkat cells with RAGE siRNA.** (a) 48 hours after transfection with the indicated siRNAs, Jurkat cells were stimulated with cross linked-OKT3 mAb for the indicated time and the levels of phosphorylated and total ZAP70 were determined by Western blot. The expression of pZAP70 were reduced in the RAGE siRNA treated cells. (n = 3 replicate studies, p = 0.0002 by a two-way ANOVA and the effects of the siRNAs were compared.) (b) A representative blot of total and pZAP70 are shown.

inflammatory phenotype that can be found in patients with type 1 and type 2 diabetes [21, 22]. There are many ligands for RAGE which are increased in several pathologic states ranging from diabetes and other autoimmune conditions to vascular disease [11, 23–30]. Moreover, there are abundant ligands available for RAGE that is expressed in the endosomes from extracellular sources as well as intracellular HMGB1 [31]. Furthermore a number of different mechanisms may account for RAGE/ligand interactions. For example, through engagement of TLR9 that is expressed in the endosomes, DNA from necrotic cells may be made available to RAGE that is colocalized [11]. Thus, the expression of RAGE itself, rather than ligands are most likely to be a limiting step in affecting T cell activation.

There are limitations to our studies. First, the responses in Jurkat cells may not completely reflect the activation of primary human T cells. Other cellular responses, such as cytokine

responses or even proliferation are known to be different between Jurkat and primary cells. It is possible that there are off target effects from the CRISPR deletion that may have affected cellular functions, but the confirmation of the effects of RAGE silencing together with observations from clinical samples suggest this is not the explanation for our findings. In addition, the precise mechanism whereby RAGE modulates signaling molecules and the precise target(s) of RAGE interactions remain unclear, but does not reflect a global defect in the cells.

In summary, we have shown that RAGE is involved in early events in T cell signaling. The multiple ligands that are available suggest that this mechanism is important in modulating the amplitude of adaptive T cell responses.

## Supporting information

**S1 Fig. Analysis of RAGE/KO Jurkat cells.** The primer pairs used for RAGE knockout detection were forward 5'-TGTTCCCCAGCCTTGCCTTCAT-3' and reverse 5'-GCCCCTCC TCGCCTGGTTCT-3', which generate a 533 bp PCR product unique to the RAGE gene deletion. Primers that were specific for the WT or excised fragment were prepared and the genomic DNA was analyzed by PCR. The absence of RAGE expression was confirmed by DNA amplification and FACS.
(PDF)

**S2 Fig. CD3 expression in WT and RAGE KO Jurkat cells.** a: the frequency of Jurkat WT and KO cells that are positive for CD3 are shown (Student's t-test, ***$p < 0.001$, n = 8,6) b: The CD3 mean fluorescence index is shown (Student's t-test, *$p < 0.05$, n = 11,7), c: an example of staining. The filled bar shows the fluorescence of isotype control, the solid line shows the fluorescence of CD3 on the KO and the dotted line on the WT cells.
(PDF)

**S1 Raw image. RAGE Gel.**
(PDF)

**S1 Data. Underlying figure data.**
(ZIP)

## Acknowledgments

Special thanks to Peter Rabinovich of Yale Department of Pathology for transfection protocols.

## Author Contributions

**Conceptualization:** Paula Preston-Hurlburt, Kevan C. Herold.

**Data curation:** James C. Reed, Paula Preston-Hurlburt, Kevan C. Herold.

**Formal analysis:** James C. Reed, Carrie Lucas, Kevan C. Herold.

**Funding acquisition:** Kevan C. Herold.

**Investigation:** James C. Reed, Paula Preston-Hurlburt, William Philbrick, Gabriel Betancur, Kevan C. Herold.

**Methodology:** William Philbrick, Maria Korah, Carrie Lucas, Kevan C. Herold.

**Project administration:** Kevan C. Herold.

**Supervision:** Paula Preston-Hurlburt.

**Writing – original draft:** James C. Reed, Kevan C. Herold.

**Writing – review & editing:** James C. Reed, Paula Preston-Hurlburt, William Philbrick, Carrie Lucas, Kevan C. Herold.

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
