## [Decision Letter · Decision Letter 0]

11 Jun 2020

PONE-D-20-15463

The Receptor for Advanced Glycation Endproducts (RAGE) modulates T cell signaling

PLOS ONE

Dear Dr. Herold,

Thank you for submitting your manuscript to PLOS ONE. After careful consideration, we feel that it has merit but does not fully meet PLOS ONE’s publication criteria as it currently stands. Therefore, we invite you to submit a revised version of the manuscript that addresses the points raised during the review process.

We look forward to receiving your revised manuscript.

Kind regards,

Matthias G von Herrath, MD PhD

Academic Editor

PLOS ONE

Additional Editor Comments:

It would be good to evaluate more donors, if possible, or, if that is not feasible, conclusions would have to be tempered by quite a bit, I'd suggest. Preference would be to look at more donors and also pay attention to the other issues raised.

Journal Requirements:

2. In your Methods section, please provide additional details regarding the cell lines used in your study. Please include the source from which you obtained the cells, the catalog numbers if applicable, whether the cell lines were verified, and if so, how they were verified. For more information on PLOS ONE's guidelines for research using cell lines, see https://journals.plos.org/plosone/s/submission-guidelines#loc-cell-lines.

3. PLOS ONE requires experimental and statistical methods to be described in enough detail to allow suitably skilled investigators to fully replicate and evaluate a study. See (1) https://journals.plos.org/plosone/s/submission-guidelines#loc-materials-and-methods and (2) https://journals.plos.org/plosone/s/submission-guidelines#loc-statistical-reporting for more information.

To comply with PLOS ONE submission guidelines, in your Methods section, please provide additional information regarding your methodology and statistical analyses.

In addition, please ensure that you describe the sources and catalog numbers (if applicable) of all cell lines, reagents, proteins, antibodies, equipment, etc. in the methods section of your manuscript. For antibodies, please include dilutions or final concentrations used in the experiments.

4.PLOS ONE now requires that authors provide the original uncropped and unadjusted images underlying all blot or gel results reported in a submission’s figures or Supporting Information files. This policy and the journal’s other requirements for blot/gel reporting and figure preparation are described in detail at https://journals.plos.org/plosone/s/figures#loc-blot-and-gel-reporting-requirements and https://journals.plos.org/plosone/s/figures#loc-preparing-figures-from-image-files. When you submit your revised manuscript, please ensure that your figures adhere fully to these guidelines and provide the original underlying images for all blot or gel data reported in your submission. See the following link for instructions on providing the original image data: https://journals.plos.org/plosone/s/figures#loc-original-images-for-blots-and-gels.

Reviewers' comments:

Reviewer's Responses to Questions

**Comments to the Author**

1. Is the manuscript technically sound, and do the data support the conclusions?

Reviewer #1: Partly

2. Has the statistical analysis been performed appropriately and rigorously? 

Reviewer #1: Yes

3. Have the authors made all data underlying the findings in their manuscript fully available?

Reviewer #1: Yes

4. Is the manuscript presented in an intelligible fashion and written in standard English?

Reviewer #1: Yes

5. Review Comments to the Author

Reviewer #1: Thank you for submitting your manuscript. In its current form, I can't support publication. The data is limited to primary samples from 3 donors with T1D & 3 donors with no history of disease along with cell line-based experiments. The number of samples evaluated is too low to conclude very much, reliably. Additional samples are needed as is some input regarding T-cell subsets--even basic breakdowns between naive, effector, & memory would be informative. Evaluation of function of the primary cells using an antigenic stimulation, possibly a virus antigen pool vs. T1D-relevant peptides in addition to the anti-CD3 stimulation should also be performed to support your conclusions. Have you or do you plan to use the siRNA with PBMC that was used with the Jurkat?

Other minor issues to be addressed:

* Minimal donor information should be provided -- age & sex, at least?

* Line 89, 95, 96, etc. – lower case ‘u’ is used in place of mu symbol—please correct all instances throughout the manuscript

* Sentence on line#91-92 is poorly written & should be modified.

* Tenses are inconsistent throughout the manuscript. In general, methods sections should be past tense.

* Line 217 – space between ‘without’ and ‘anti-CD28’

6. PLOS authors have the option to publish the peer review history of their article (what does this mean?). If published, this will include your full peer review and any attached files.

Reviewer #1: No

---

## [Author Response · Author response to Decision Letter 0]

26 Jun 2020

Responses to review PONE-D-20-15463

Response to Associate Editor

We agree with the importance of having a sufficient number of subjects to avoid type 1 and type 2 errors. We note that our previous version of the manuscript had an error in the numbers of patients analyzed – the correct numbers are 8 patients (not 3) with T1D and 6 healthy control subjects. The data from these patients and controls are shown in Figure 1. 

We felt that it was important to give additional background on the studies that led to these investigations in T cell signaling as we have previously published information from 30 patients with T1D (including 7 in our PlosOne paper (Akirav et al PLoS One. 2012;7(4):e34698) and 23 patients at risk for T1D. The latter group, which involved participants in a TrialNet study, 9 progressed to overt T1D and 13 did not. We reported increased expression of RAGE in T cells prior to onset of T1D. Because of the extensive number of previously reported cases and characterization of the RAGE+ T cells, we did not feel that showing the frequency of RAGE in T cells from patients with T1D was again needed. 

 

Journal Requirements

2. We have clarified the Jurkat cells that were used. 

3. We have revised the Methods section as requested. 

4. We have provided the uncropped gel that was used to generate Supplementary figure 1

5. We have removed the phrase “data not shown”. It refers to the transfection of the Jurkat cells with the commercially available RAGE siRNAs. It is a technical matter and we feel that that text description would be sufficient. 

 

Reviewer 1

1. Thank you for submitting your manuscript. In its current form, I can't support

publication. The data is limited to primary samples from 3 donors with T1D & 3 donors with

no history of disease along with cell line-based experiments. 

Response: Please see our response to the Editor. The manuscript includes data on 8 patients and 6 healthy control subjects. (Now with additional information about the patients.) This is in addition to the 52 other patients or at-risk subjects whom we have studied. In addition, in our previous publication (Durning et al J Immunol 2016), we described the phenotype of the RAGE+ CD4+ and CD8+ cells. They are memory cells that also express EOMES, KLRG1, IRF4, and CXCR3. 

In addition to the above characterization, we have previously studied RAGE expression on viral antigen reactive cells (Akirav PlosOne 2012) and found expression on the antigen reactive cells (identified by Class I MHC tetramers) with increased expression when they were cultured with antigen. 

We have added new text to the Background to review this information. 

2. Have you or do you plan to use the siRNA with PBMC that was used with the Jurkat?

Response: For technical reasons, associated with silencing with siRNA in primary human T cells we chose instead to differentiate responses in RAGE+ and RAGE- human T cells, which is shown in Figure 1. The technical issues not only involve the difficulties in silencing in primary cells but also the need to activate and expand the primary cells prior to transfection with the siRNA. The activation and expansion itself enhances RAGE expression. Jurkat cells are frequently used for studies of T cell signaling. We felt, therefore, that studies in this cell line were relevant to human cells and our data supports that the lessons from the cell line are applicable. 

3. Other minor issues to be addressed:

* Minimal donor information should be provided -- age & sex, at least?

Response: thank you for this suggestion. We have now included this. 

4. * Line 89, 95, 96, etc. – lower case ‘u’ is used in place of mu symbol—please correct all

instances throughout the manuscript

Response: we corrected this error. Thanks for pointing it out. 

5. * Sentence on line#91-92 is poorly written & should be modified.

Response: We agree and changed the wording. 

6. * Tenses are inconsistent throughout the manuscript. In general, methods sections should be past tense.

Response: We have reworded the methods section and clarified a number of the sentences. 

7. * Line 217 – space between ‘without’ and ‘anti-CD28’

Response: we corrected this error.

---

## [Decision Letter · Decision Letter 1]

10 Jul 2020

PONE-D-20-15463R1

The Receptor for Advanced Glycation Endproducts (RAGE) modulates T cell signaling

PLOS ONE

Dear Dr. Herold,

Thank you for submitting your manuscript to PLOS ONE. After careful consideration, we feel that it has merit but does not fully meet PLOS ONE’s publication criteria as it currently stands. Therefore, we invite you to submit a revised version of the manuscript that addresses the points raised during the review process.

We look forward to receiving your revised manuscript.

Kind regards,

Matthias G von Herrath, MD PhD

Academic Editor

PLOS ONE

Reviewers' comments:

Reviewer's Responses to Questions

**Comments to the Author**

1. If the authors have adequately addressed your comments raised in a previous round of review and you feel that this manuscript is now acceptable for publication, you may indicate that here to bypass the “Comments to the Author” section, enter your conflict of interest statement in the “Confidential to Editor” section, and submit your "Accept" recommendation.

Reviewer #1: All comments have been addressed

2. Is the manuscript technically sound, and do the data support the conclusions?

Reviewer #1: Yes

3. Has the statistical analysis been performed appropriately and rigorously? 

Reviewer #1: Yes

4. Have the authors made all data underlying the findings in their manuscript fully available?

Reviewer #1: Yes

5. Is the manuscript presented in an intelligible fashion and written in standard English?

Reviewer #1: Yes

6. Review Comments to the Author

Reviewer #1: Thank you for the additional information. After the minor items are addressed below, I recommend publication.

Minor edits/questions:

* For the methods, a brief description of the cryopreservation should be added.

* line 89-90 -- authors state that they have 8 participants with T1D but the demographic description says 2 M & 5 F, they are missing 1 participant in their count. Additionally, no information regarding the healthy subjects--were they similar in sex distribution & age?

* Were the PBMC rested for any time post-thaw before culture & stimulation?

* line 166 -- author states they studied T-cells from healthy & donors with T1D and it was n = 3 each -- why weren't all donors evaluation? Or is this a typo? The figure description list n = 8 & 6 for T1D & healthy, respectively.

7. PLOS authors have the option to publish the peer review history of their article (what does this mean?). If published, this will include your full peer review and any attached files.

Reviewer #1: No

---

## [Author Response · Author response to Decision Letter 1]

15 Jul 2020

Please see the uploaded Response to Reviewers.

---

## [Editor Report · Decision Letter 2]

17 Jul 2020

The Receptor for Advanced Glycation Endproducts (RAGE) modulates T cell signaling

PONE-D-20-15463R2

Dear Dr. Herold,

We’re pleased to inform you that your manuscript has been judged scientifically suitable for publication and will be formally accepted for publication once it meets all outstanding technical requirements.

Kind regards,

Matthias G von Herrath, MD PhD

Academic Editor

PLOS ONE

Additional Editor Comments (optional):

None
---

## [Editor Report · Acceptance letter]

23 Jul 2020

PONE-D-20-15463R2 

The Receptor for Advanced Glycation Endproducts (RAGE) modulates T cell signaling 

Dear Dr. Herold:

I'm pleased to inform you that your manuscript has been deemed suitable for publication in PLOS ONE. Congratulations! Your manuscript is now with our production department. 

Kind regards, 

on behalf of

Prof. Matthias G von Herrath 

Academic Editor

PLOS ONE